Ten quick tips for electrocardiogram (ECG) signal processing

http://orcid.org/0000-0001-9655-7142 Chicco Davide 1 2 davide.chicco@gmail.com
Karaiskou Angeliki-Ilektra 3
De Vos Maarten 3 4
1 Dipartimento di Informatica Sistemistica e Comunicazione, Università di Milano-Bicocca , Milan , Italy
2 Institute of Health Policy Management and Evaluation, University of Toronto , Toronto, Ontario , Canada
3 STADIUS Center for Dynamical Systems Signal Processing and Data Analytics, Katholieke Universiteit Leuven , Leuven , Belgium
4 Department of Electrical Engineering, Katholieke Universiteit Leuven , Leuven , Belgium
Balas Valentina Emilia
Electronic publication date: 2024 Sep 3
Publication date: 2024
Volume: 10
Electronic Location ID: e2295
Received 2024 May 22; Accepted 2024 Aug 9
Copyright: © 2024 Chicco et al.
Copyright year: 2024
Copyright holder: Chicco et al.
License: This is an open access article distributed under the terms of the Creative Commons Attribution License, which permits unrestricted use, distribution, reproduction and adaptation in any medium and for any purpose provided that it is properly attributed. For attribution, the original author(s), title, publication source (PeerJ Computer Science) and either DOI or URL of the article must be cited.
License URL: https://creativecommons.org/licenses/by/4.0/

Keywords: ECG, Electrocardiography, Signal processing, Medical signal processing, Cardiology, Quick tips, Guidelines

Funding: Ministero dell’Università e della Ricerca of Italy Dipartimento di Informatica Sistemistica e Comunicazione at Università di Milano-Bicocca The work of Davide Chicco was funded by the European Union—Next Generation EU programme, in the context of The National Recovery and Resilience Plan, Investment Partenariato Esteso PE8 “Conseguenze e sfide dell’invecchiamento”, Project Age-It (Ageing Well in an Ageing Society), and was supported by Ministero dell’Università e della Ricerca of Italy under the “Dipartimenti di Eccellenza 2023–2027” ReGAInS grant assigned to Dipartimento di Informatica Sistemistica e Comunicazione at Università di Milano-Bicocca. There was no additional external funding received for this study. The funders had no role in study design, data collection and analysis, decision to publish, or preparation of the manuscript.

==============================
The electrocardiogram (ECG) is a powerful tool to measure the electrical activity of the heart, and the analysis of its data can be useful to assess the patient’s health. In particular, the computational analysis of electrocardiogram data, also called ECG signal processing, can reveal specific patterns or heart cycle trends which otherwise would be unnoticeable by medical experts. When performing ECG signal processing, however, it is easy to make mistakes and generate inflated, overoptimistic, or misleading results, which can lead to wrong diagnoses or prognoses and, in turn, could even contribute to bad medical decisions, damaging the health of the patient. Therefore, to avoid common mistakes and bad practices, we present here ten easy guidelines to follow when analyzing electrocardiogram data computationally. Our ten recommendations, written in a simple way, can be useful to anyone performing a computational study based on ECG data and eventually lead to better, more robust medical results.

Introduction

An electrocardiogram (ECG) is a time-based recording of the electrical activity of the heart through several through repeated cardiac cycles (heartbeats). The ECG signal is usually recorded through electrodes put on the skin of a patient, and its representation shows the electrical action potential measured in volts on the y axis over time on the x axis. The most common ECG device today is the 12-lead ECG, which has twelve leads that record the signals of the heart (Sörnmo & Laguna, 2006).

The electrocardiogram (ECG) is a special case of electrogram, and should not be confused with the electroencephalogram (EEG), which is the recording of the electrical activity of the brain. The ECG recording can be interpreted manually by a physician or a medical doctor, who can notice some anomalies that could indicate potential pathological conditions. With the spread of computer sciences in the biomedical domain, researchers have started to analyze ECG data through computational algorithms and ad hoc software programs able to reveal heart trends and information that otherwise would be unnoticeable by medical doctors.

This scientific area, mainly developed within the biomedical engineering community, gained the broad name of ECG signal processing. Since the 1980s, several computational projects for the analysis of electrocardiogram data have appeared, both from hi-tech companies and from academic scientific research (Berkaya et al., 2018; Clifford, Azuaje & McSharry, 2006; Gupta et al., 2023a, 2023b, 2023c; Gupta, Mittal & Mittal, 2022). Even if performing computational analyzes of ECG has become easier for software developers, making mistakes during these analyses has become easier, too. We therefore decided to present these quick tips to avoid common errors and bad practices during a computational study on electrocardiogram signals.

To the best of our knowledge, no study reporting guidelines on ECG data analysis exists in the literature nowadays. The scientific literature includes articles of recommendations and guidelines on several topics but none on ECG signal processing. We fill this gap by presenting here our recommendations: even if we designed originally these guidelines for apprentices and beginners, we believe they should followed by experts, too.

Tip 1: before starting, study how the electrocardiogram works, what it says and what it does not say

This first guideline might seem trivial and straightforward to our readership, but it actually refers to an underrated aspect of electrocardiogram (ECG) analyses. Before turning on your computer to work on the computational analyses of electrocardiogram signals, we advise studying and understanding the electrocardiogram signal first.

In fact, the electrocardiogram signal has a unique data type and structure that requires specific training. Keep that in mind. Before starting an ECG signal processing analysis, learn the main concepts of an electrocardiogram, such as P wave, Q wave, R wave, S wave, J point, T wave, QRS complex, PR segment, ST segment, PR interval, QT interval, J point, corrected QT interval, and others Fig. 1. Learn how to read an electrocardiogram by yourself with your own eyes, before turning the computer on. Then, only when you are able to understand what an ECG says and what it does not say, turn on your computer and work on your computational analysis.

Figure 1 Schematic example of an electrocardiogram (ECG).

Image publicly available under the Creative Commons Attribution 3.0 Unported (CC BY 3.0 DEED) license on Wikimedia Commons (CFCF, 2023). We advise learning and understanding the main concepts of an electrocardiogram (for example, P wave, Q wave, R wave, S wave, J point, T wave, QRS complex, PR segment, ST segment, PR interval, QT interval, J point and corrected QT interval) before performing the computational analysis. Would you be able to explain this image to first person around you?

Multiple resources to study the ECG are available online, including several free educational videoclips on YouTube. We recommend studying the content of ECGpedia (De Jong, 2023), an online encyclopedia similar to Wikipedia, started by Jonas de Jong and fully dedicated to the electrocardiogram theme.

And how do you know if you understood the meanings of the ECG well enough? Explain it to someone who has no medical knowledge, and see if they comprehend. The mathematician David Hilbert once said: “A mathematical theory is not to be considered complete until you have made it so clear that you can explain it to the first man whom you meet on the street”. That also applies to the ECG understanding.

Tip 2: understand the medical context and the medical problem you are trying to solve

Once you have clearly apprehended what the ECG says, it may be tempting to dive directly into your computational tools, but before starting this, you must clearly define your scientific question and be sure what your aim and potential pitfalls are. Furthermore, given the enormous number of medical data, it can be overwhelming to understand what is valuable information and what is a nuisance. Multiple factors can affect the data, and without the appropriate training, it is difficult to decipher if there is a problem with your methodology, if the medical context of your problem limits you, or if it is influenced by the data acquired from a population with specific characteristics. Naturally, this also applies in the case of ECG data. Thus, now it is time to focus on understanding the medical context and problem you are facing.

Before all else, you should learn how to interpret the electrocardiogram (ECG) signals you see based on the problem you are trying to solve. Medical doctors and physicians have the experience to distinguish with a bare eye not only the abnormalities of the ECG but also to give a more specific diagnosis of what these abnormalities represent. However, do not rely only on experts’ labels; even experienced medical experts provide contradicting diagnoses. Therefore, you should look into these abnormalities and learn how to extract patterns so that when you apply your algorithms, you are sure that what you captured is what you expected to catch. Take, for example, the medical problem of arrhythmia, where there are multiple categories with diverse characteristic manifestations, excessively slow or fast heartbeats such as sinus bradycardia and atrial tachycardia, irregular rhythm with missing or distorted wave segments and intervals, or both (Zheng et al., 2020). If your scientific problem involves a specific target group, such as athletes (Drezner, 2012), then focus on the ECG abnormalities relevant to this particular population. Additionally, on the assumption that you work with 12-lead ECG signal, pay attention to understanding what kind of information can be extracted and interpreted from each lead (Garcia, 2015).

Nevertheless, to master the electrocardiogram (ECG), collaboration with medical experts is key. Discuss your scientific problem with your clinicians, including cardiologists or medical doctors, to define and assess your goals from the beginning. Consistently share your results with them to evaluate and review them. Optimally, seek experts close to your workplace so that you can easily access their valuable knowledge; however, if this is impossible, consider connecting with healthcare professionals online. In that way, you are assured that your scientific goals stay relevant.

That being said, pay attention to the scientific breakthroughs relevant to the scientific problem you are working on. Dive into the literature and be informed of the most representative features and models used in previous works relevant to your medical problem. Then, try to reproduce those results. There might be a chance that your ECG entangles information for various medical problems, and therefore, you should be careful about what datasets you are going to use, as we see in the next tip.

Tip 3: understand if you have data suitable for making progress on the scientific problem you are investigating

After gaining a solid grasp of the medical context and issue at hand, the next step is the selection of the appropriate dataset to address your scientific question. Understanding the suitability of data is a ground step in any scientific study, and it is no different when it comes to electrocardiogram (ECG) signal processing. The relevance and quality of the data you work with significantly impact the progress and success of your scientific work.

Above all else, your data must be directly relevant to the scientific question you are trying to solve. ECG signal studies typically focus on specific health issues, conditions, states, or populations. Being assured that the data you are going to analyze were collected from the appropriate medical context is of very high importance in order to achieve meaningful results. Irrelevant data can lead to inaccurate conclusions and wasted research efforts.

The quality of your data is of equal importance. You should be able to understand the quality of your signal and ensure that what you see is not an artifact (Fig. 2). Once again, this is directly connected with the medical context of the problem that you are facing. Was the ECG acquired during movement, sleep, or during a 24-hours span (Moeyersons et al., 2019; Pérez-Riera et al., 2018; Carr et al., 2018; Carr, de Vos & Saunders, 2018; Gregg et al., 2008; Clifford, 2006)? Comprehending this information and its effect on your electrocardiogram (ECG) data will save you a lot of time and energy when you are trying to find a solution. The assessment of ECG signal quality is an active topic of research, and various quality indices have been developed for this reason, as we will see later (Tip 5).

Figure 2 ECG Segments contaminated by artifacts.

Image taken from Varon, van Huffel & Suykens (2015) with the authors’ permission. (A) In grey the original electrocardiogram (ECG) contaminated by power line interference at 50 hertz is indicated, and in black its filtered version. (B) Contact noise. (C) Electrode motion. (D) Muscle artifact.

Data quantity is also critical. An insufficient amount of data can result in statistical insignificance, limiting the complexity of your analyses and the depth of your insights. A large dataset enhances the statistical power of your research, without falling into the p-hacking trap (Chén et al., 2023; Makin & de Xivry, 2019). The length of the data you are using must also be taken into account. Do you want to perform peak detection? Usually, the algorithms suggested in the literature require at least 10 seconds of continuous ECG recordings to have relatively robust results. Still, if you have longer windows, this would be much more beneficial. Is your aim to study the heart rate variability (Clifford, 2002; Ernst, 2014)? Then, you should possess recordings of at least 3 minutes of ECG length. Moreover, the time you record data can change the variability numbers a lot. So, remember to avoid comparing heart rate variability numbers from different time lengths (Cygankiewicz & Zareba, 2013).

Last but not least, you should pay attention to the ECG’s sampling rate since it can seriously affect various measures derived by the electrocardiogram (ECG) signal analysis. Reduced sampling rates can compromise the precision of detecting R-wave fiducial points and, therefore, greatly affect your Heart Rate Variability (HRV) analysis (Garcia-Gonzalez, Fernandez-Chimeno & Ramos-Castro, 2004; Mahdiani et al., 2015; Ellis et al., 2015; Petelczyc et al., 2020; Lu et al., 2023). In fact, for clinical applications, the ECG sampling frequency that is nowadays recommended is at least 512 hertz (Hz) (Petelczyc et al., 2020). Besides, the population on which your medical context is based plays an essential role in deciding the optimal sampling frequency. For instance, if your research involves the analysis of fetal ECG, it is recommended that a good dataset should be acquired with a sampling frequency of 2 kHz and resolution of 16 bits (Sameni & Clifford, 2010).

Tip 4: use only open source programming languages and software platforms, if sufficient for your scientific project

When one starts working on a computational project, they have to face an important question: which programming language should I use? Our recommendation for electrocardiogram (ECG) analysis is the same we already provided in other studies: choose an open source programming language such as Python, R, or GNU Octave, if their functionalities are sufficient for your scientific goals.

Employing open source code, in fact, would bring multiple advantages. First, open source programming languages are free, and one does not need to spend their money or their institution’s money to purchase a license, as it would happen for proprietary software. Second, an open source scripts can always be shared freely and openly among colleagues and collaborators. If one needs to send their software scripts to other people so that they could use them on their computers, only open source programs would give this possibility. On the contrary, if the ECG analysis was performed with proprietary software scripts, they could be shared and used only with other developers who have a license to that proprietary programming language.

Third, open source programs can be used and re-used by a developer even in case they changed laboratories, institutes, or companies. If one develops an ECG analysis script in Python, for example, and then moved to another scientific laboratory, they would still be able to re-use their own script in the new work environment. On the contrary, if they developed their script in an proprietary programming language, and the new work environment did not have the license for their usage, they would be unable to re-use it and would need to re-implement it.

It is also relevant to mention that currently, Python is the most popular programming language in the world, according to the PYPL index (PYPL, 2023), to the TIOBE index of November 2023 (TIOBE, 2023) and the Kaggle 2022 survey (Kaggle, 2022). R has excellent ranks in these standings as well: 7th most employed programming language worldwide in November 2023 for the PYPL index (PYPL, 2023), 19th at the same date for the TIOBE index (TIOBE, 2023), and 3rd in the Kaggle 2022 survey (Kaggle, 2022). Therefore, if a student or an apprentice coder decided to invest their time and energy to learn Python or R for ECG signal processing, it would be likely that they could take advantage of their gained skills in other projects as well.

Both Python and R provide several software packages for electrocardiogram signal processing. Python furnishes the popular HeartPy (Van Gent et al., 2019; Van Gent et al., 2018) and other software libraries such as ecgtools (pypi ecgtools, 2024), ecg-quality (pypi ecg-quality, 2023), ndx-ecg (pypi ndx-ecg, 2023), PhysioZoo (PhysioZoo, 2023; Behar et al., 2023), pyPPG (Goda, Charlton & Behar, 2024), and ecghelper (pypi ecghelper, 2023), while R supplies RHRV (Martínez et al., 2017; The Comprehensive R Archive Network, 2023).

Of course, we recommend the usage of open source programming languages only if their libraries and functionalities are sufficient and suitable to solve the scientific problem analyzed.

If a user understood that a particular analysis on electrocardiogram data would be possible only with proprietary software, they should go for it. For all the other cases, we advise to choose open source.

Tip 5: take care of noise and of baseline wander

Broadly speaking, biomedical datasets for computational analyses can belong to two types: ready to be used and unready to be used. Usually, datasets containing data from electronic health records (EHRs) of patients with a particular disease, collected within a hospital, can be considered ready to be used for a computational analysis. These datasets often contain data from laboratory tests and need no preprocessing before usage (Chicco & Jurman, 2020a). However, for datasets of other types need an effective preprocessing step on the raw data before being ready to undergo a computational analysis. For example, medical images’ data need noise removal (Chicco & Shiradkar, 2023), while microarray and RNA-seq gene expression data need batch correction (Sprang, Andrade-Navarro & Fontaine, 2022; Chen et al., 2011).

Electrocardiogram data belong to this second category: you need to remove noise before doing your scientific analysis on these data. Several filtering methods to remove electrocardiogram (ECG) noise exist in the scientific literature (Sulthana, Rahman & Mirza, 2018; Salman, Rao & Rahman, 2018; Castroflorio et al., 2013; Zhang et al., 2014; Chui et al., 2016; Nejadgholi, Moradi & Abdolali, 2011; Afkhami, Azarnia & Tinati, 2016; Nuthalapati et al., 2019; Li, Rajagopalan & Clifford, 2013). ECG data also suffer from baseline wander, which is an artifact noise signal generated by the movements of the patients, including breathing. Moreover, peripheral or implanted devices can also generate those artifacts (Fig. 2). ECG data therefore necessitate baseline wander correction (Blanco-Velasco, Weng & Barner, 2008; Leski & Henzel, 2005; Kaur, Singh & Seema, 2011; Zhang, 2006; Sörnmo, 1993; Jane et al., 1992) as one of the first preprocessing step.

Nevertheless, removing the baseline wonder from the ECG does not guarantee that the signal is now ready for analysis. Thus, the use of multiple quality metrics for the electrocardiogram (ECG) is also essential. Various methods have been suggested to assess the quality of ECG recordings automatically. These methods, known as Signal Quality Indices (SQIs), measure the level of interference and evaluate the signal’s suitability for its intended purpose.

Some of the benchmark ECG quality metrics that should always be used are the kurtosis of the signal, where larger values of this metric are associated with higher quality, and skewness, where ECG is expected to be highly skewed because of the QRS complex, and therefore, lower values would indicate lower quality of ECG (He, Clifford & Tarassenko, 2006; Seeuws, De Vos & Bertrand, 2021; Behar et al., 2023). Similarly, metrics extracted from the frequency domain are of equal significance. Those include the in-band (that means, 5–40 Hz) to out-of-band spectral power ratio (IOR) within the QRS complex, the relative power in the QRS complex (that means, 5–15 Hz), and the relative power in the baseline (that means, frequency content below 1 Hz) where in all those metrics, the larger their value, the higher the ECG signal quality (Tobón, Falk & Maier, 2014; Seeuws, De Vos & Bertrand, 2021).

Moreover, metrics depending on the beat detection should also be considered. Those include the rSQI metric, which is the ratio of the number of beats detected by the WQRS algorithm (Zong, Moody & Jiang, 2003) to the number of beats detected by the Epilimited algorithm (Hamilton, 2002), as well as the bSQI metric which is the ratio of beats detected by the WQRS that matched with beats detected by the Epilimited, where again in both metrics the higher the value indicates higher quality of the ECG signal (Behar et al., 2023; Seeuws, De Vos & Bertrand, 2021).

In a nutshell, we recommend you never analyze raw electrocardiogram data as they come from electrocardiogram machines. Always preprocess them, taking care of noise and of baseline wander. Similarly, remember to use SQIs to assess the quality and suitability of your data for the scientific problem you are facing. You can proceed with your computational analysis only when you know your data is adequately preprocessed and suited in terms of quality for your medical problem.

Tip 6: start with the simplest methods, and use complex methods only if it is necessary

Common analyses of electrocardiogram signals often include the feature extraction of temporal and morphological variables (Ardeti et al., 2023). To this end, researchers employed traditional electrocardiogram (ECG) signal processing algorithms such as several finite impulse response (FIR) (Lastre-Dominguez et al., 2019), derivative (Lastre-Dominguez et al., 2019), windowing (Oktivasari et al., 2019) and transformation-based algorithms (Fujita, Sato & Kawarasaki, 2015) in the past.

Especially with the spread of machine learning and deep learning, researchers analyzing ECG signals might be tempted to use more complex methods just because they are more trendy or in fashion. Instead, we recommend to start any analysis by utilizing simple algorithms, chosen among the traditional ones; they are after as good as complex ones (Andreotti et al., 2017). Do not start with complex stuff.

If the simple, traditional algorithms applied to the ECG data are sufficient to solve your scientific problem, just stick with them. Only if no simple computational technique can do it, move to more complex methods, such as machine learning and deep learning. Using simple methods will give you the chance to keep everything under control and to see results generated in an interpretable way, which is pivotal especially for the clinicians that will use your results (Rudin, 2019).

Along these lines, we suggest to start your preliminary analysis by using open source toolboxes or software libraries created specifically for dealing with ECG analysis (Van Gent et al., 2019; Martínez et al., 2017). In fact, other researchers have developed those libraries which were tested thoroughly before publication. Examples of such libraries are the already-mentioned HeartPy (Van Gent et al., 2019, 2018) package in Python and the RHRV (Martínez et al., 2017; The Comprehensive R Archive Network, 2023) package in R, both open source and publicly available for free. Other software repositories on electrocardiogram signal processing are available on GitHub (2023).

Tip 7: choose the most suitable metrics for your scientific problem and never rely on a single one

When you have decided on the methodology you are going to follow, how will you know if it is actually good enough for your problem? Take, for instance, the task of heartbeat classification, a problem studied for decades but still needs to be solved. The problem of heartbeat classification is affected by the small number of training samples, data imbalance both in the number of samples for each class and the number of distinct features for each class, the large subject variability of the ECG signals, as well as the morphological similarity between different beats (Wang, Zhou & Yang, 2022; Jiang et al., 2019; Villa et al., 2019). Thus, employing a performance metric like accuracy would lead to misleading results. For a problem like this, it would be wise to use various metrics that consider the imbalance between the classes, the imbalance between the different features, and the variability between the subjects. That way, assessing and comparing different algorithms or methods would be more straightforward. Therefore, we recommend providing multiple performance metrics, like the ones mentioned later in this section, and good arguments on why each performance metric was chosen and what it represents (Chicco & Shiradkar, 2023).

Accuracy is the simplest and, thus, most typical way to evaluate a method performing ECG peak detection, beat or signal classification, clustering, and regression (Jambukia, Dabhi & Prajapati, 2015; Joloudari et al., 2022; Nezamabadi et al., 2023). However, accuracy is a measure that only shows the amount of correctly predicted classes and can be proven highly deceptive for datasets with unbalanced classes. For example, when you have thousands of hours of ECG recordings, and you create a model based on these that predicts an event that only occurs in a small portion of time relative to the whole duration of the recordings, you will have a high accuracy performance since the amount of the actual events is insignificant compared to the whole amount of your available data. Thus, it will not affect the overall accuracy even if it is not predicted.

For that reason, it is highly advised always to avoid accuracy and to include more metrics that account for imbalanced datasets while also considering the importance of each metric for each scientific problem. When dealing with a classification problem, always include sensitivity, specificity, precision, and negative predictive value in your reported results. Moreover, remember to include a metric that considers the whole confusion matrix for evaluating your model, the Matthews correlation coefficient (MCC) (Jurman, Riccadonna & Furlanello, 2012; Chicco, 2017; Chicco & Jurman, 2020b; Chicco & Jurman, 2023a, 2023b, 2022; Chicco, Starovoitov & Jurman, 2021; Chicco, Tötsch & Jurman, 2021) metric, which will clearly indicate whether your model performs well in all the corresponding classes. An alternative to MCC is the Kappa Cohen coefficient, which is quite famous among applications with imbalanced datasets like sleep stage detection; however, it should be used more cautiously (Delgado & Tibau, 2019; Chicco, Warrens & Jurman, 2021).

If you still want to report accuracy for interpretation reasons, consider a weighted accuracy measure (Li, Rajagopalan & Clifford, 2014), which has also been proposed as a way to describe the performance of models dealing with imbalanced datasets, though always combined with those measures as mentioned above. The j index (Soria & Martinez, 2007), which adds the sensitivity and specificity of the most important classes, has also been suggested for beat classification problems that deal with imbalanced, like arrhythmia datasets (Villa et al., 2019). For applications like seizure detection, a pivotal evaluation measure is the false detection/alarm rate per day (Jeppesen et al., 2023; Vandecasteele et al., 2017; De Cooman et al., 2017; Bhagubai et al., 2023). Further, if you aim to create a robust electrocardiogram (ECG) peak detection algorithm, you should also consider checking the error rate. Concerning regression analysis, as already mentioned in other tips (Chicco & Shiradkar, 2023), the way to proceed is by always including the coefficient of determination (R-squared or R2), symmetric mean absolute percentage error (SMAPE), mean square error (MSE), root mean square error (RMSE), mean absolute error (MAE), mean absolute percentage error (MAPE), and build the rankings of the methods on R-squared (Chicco, Warrens & Jurman, 2021; Chicco & Shiradkar, 2023).

Concerning clustering scientific problems, you should always report measures like the Davies-Bouldin index (Davies & Bouldin, 1979), the Calinski-Harabasz index (Łukasik et al., 2016), the Dunn index (Dunn, 1974), Gap statistic (Tibshirani, Walther & Hastie, 2001), and the Silhouette score (Kaufman & Rousseeuw, 2009; Chicco & Shiradkar, 2023) in combination with other metrics relevant to your research objective. Other similarity metrics which can be considered are the Jaccard coefficient and normalized mutual information (Nezamabadi et al., 2023). Nonetheless, if you already have ground truth labels of what you will expect, you can also use this information. For statistical analyses, always include the adjusted p-value (Jafari & Ansari-Pour, 2019; Chén et al., 2023), by using a 0.005 significance threshold (Benjamin et al., 2018). The take-home message here is that you should always check the recent literature to find out which metrics are used with respect to your scientific problem, use multiple metrics and not only one, and be careful that using the wrong metrics can fool you and fool the readership.

Tip 8: prepare your results in a format that can be clearly understood, interpreted, and used by the clinicians

Collaboration with medical experts in the field is one of the cornerstones of computational medical research. In electrocardiogram (ECG) research, this is not different. You have already asked their opinion concerning the medical context and defined your scientific goal with them. It is now time to present your results to either take their feedback or prepare a manuscript together. Yet, medical experts may not have the computational background to fully follow the steps your methods are based on. They also might not be aware of certain metrics you used without further explanation from your side. Therefore, you need to be sure that your results are in a format that is easily comprehended.

A picture means a thousand words. Or, for the scientific case, a figure means a thousand words. Try to create figures showing all the appropriate information and the message you want to convey. Is the message that one method works better than the other? Smartly use your predefined evaluation metrics and create figures that clearly show your novelty. If, for example, you have worked on an electrocardiogram (ECG) peak detection algorithm that can clearly distinguish the PQRST wave of ECG even in very noisy data, then visualize a noisy ECG and what your algorithm captures. If you study features that are difficult to understand by scientists without a technical background, find a way to create a self-explanatory visualization or an analogy that can easily explain what you are taking into account and what it represents. Create figures that could be directly used in a publication, both for your convenience and mostly to be sure that what you present is readable and comprehensible (Rougier, Droettboom & Bourne, 2014; Chicco & Jurman, 2023c).

Does the clinician want to carry out some further analysis? Ask them in what format they would want to see those features you extracted so they can do their own analysis afterward. For example, provide them a data frame in comma-separated values (CSV) format that includes all the information you have extracted, making abstraction of the method itself. In those tables, we suggest you include the information on all the variables relevant to the problem, like age, sex, and different health conditions related to the scientific problem as different categorical values, and finally, all the other measures you calculated during your analysis while ensuring the patients’ confidentiality. Thereon, they can use tools or software they prefer to perform their own analysis.

Finally, make sure to write a report explaining your methods and all the figures you created. Preferably, write this report as you would for a scientific paper, namely, explain thoroughly the methodology you used and the results you achieved, and add clear captions to your figures and tables (Ehrhart & Evelo, 2021). Subsequently, this report could be directly used as part of a potential manuscript draft, helping you save time and reduce the need for revisions when that moment arrives (Chicco & Jurman, 2023c).

Tip 9: look for a validation cohort dataset to confirm the results you obtained on the primary cohort dataset

After performing computational analysis on your main electrocardiogram dataset, which we can call primary cohort, you might have found interesting results that can tell you something relevant about the cardiac cycles of the patients. Even if these results might seem promising, if they were found on a single dataset, they can be considered specific to that cohort of patients, and they might lack generizability.

How can you assess if your scientific results stand not only for the patients of the dataset analyzed but rather for all the patients with the same conditions? An effective way to tackle this issue is to look for an alternative dataset of data recorded from other patients but having the same features of the primary cohort dataset. This dataset is usually called validation cohort.

Repeating your computational analysis on a validation cohort dataset would allow you to confirm or reject the discoveries made on the primary dataset. See for example this study on electronic health records of patients diagnosed with sepsis (Chicco & Jurman, 2020a).

Of course, often it is difficult to find a validation dataset compatible with the primary one, but it is always worth trying. Therefore, this is our recommendation: look for an ECG dataset of the same type of your primary dataset from the clinicians you know or online on open data repositories such as Kaggle (2023), Figshare (2023), Zenodo (2023), PhysioNet (2023), the University of California Irvine Machine Learning Repository (University of California Irvine, 2023), re3data (Registry of Research Data Repositories, 2023), or Google Dataset Search (Google, 2023). Several electrocardiogram signal datasets (such as PTB-XL (Wagner et al., 2020; Strodthoff et al., 2023), MedalCare-XL (Gillette et al., 2023), and others (Khan, Hussain & Malik, 2021; Silva et al., 2014; Zhang et al., 2015; Liu et al., 2022; Zheng et al., 2020)) have been released and published with peer-reviewed articles within journals like Scientific Data in the last years.

If you find one, repeat your computational analysis on it and include its results in your study.

Tip 10: release your data and software code publicly online and submit your article to an open access journal, if possible

In the previous tip (Tip 9), we recommended to look for an electrocardiogram (ECG) validation cohort dataset online, where to re-run your computational analysis and perhaps verify your results obtained on the primary cohort dataset. These online data repositories are useful for this scope, since they can have multiple datasets to offer for secondary analyses. However, these online data repositories should not only be exploited, but also helped to grow: since your ECG dataset could serve as a validation cohort for someone else’s study, we recommend you to openly share it online.

Of course, this sharing is possible only when you have the necessary authorization to openly share the ECG data of the patients, obtained from the ethical committee of your hospital or of your institution. If you have this authorization, we advise you to proceed and share both your raw ECG data and your ECG preprocessed data on public repositories (such as the already-mentioned Kaggle (2023), Figshare (2023), Zenodo (2023), PhysioNet (2023), or University of California Irvine Machine Learning Repository (University of California Irvine, 2023)).

Sharing your data online would make it possible to anyone around the world to re-analyze your dataset and to discover something new regarding cardiology, allowing the progress of scientific research and facilitating new scientific discoveries.

Regarding reproducibility, we recommend to make your scripts public online on GiHub or GitLab. This way, anyone will be able to re-use your computational methods, understand how you implemented them and use them, make changes to discover new scientific outcomes and verifying if you made any mistake in your analyses (Barnes, 2010). If your software code is publicly available online, your scientific discoveries can be considered more robust and transparent, since you allowed the reproducibility of your study (Markowetz, 2015).

Moreover, if you have a say on which scientific journal you and your collaborators will choose for the paper submission, we recommend you to select an open access one. If published in an open access venue, your study’s article will be available to anyone in the world, and not only to members of academic institutions or to members of high-tech companies which pay restricted-access journals to read their articles. Open access articles, in fact, can be read for free even by people in the least developed countries, currently facing wars (in Somalia and South Sudan, for example), and by high schools students who would like to delve into a specific theme. A list of health informatics open access journals where to submit an article on electrocardiogram data can be found on ScimagoJR (Scimago Journal Ranking, 2023).

Conclusions

Electrocardiogram (ECG) is a powerful and useful tool to understand the health of a heart: anomalies caught by an ECG can mean the difference between life and death for a patient. The precious information provided by an electrocardiogram can lead to important decisions on the patient’s health, such as the need to have a surgery or to start a new therapy.

Computational analyses of ECG signal, usually called ECG signal processing, can provide additional insights and pieces of information about the patient’s health to the medical doctors. Results of ECG signal processing, in fact, can highlight trends and outcomes that would otherwise be impossible to noticed by the physicians. We prepared our tips by considering only ECG data acquired digitally, but there is also an older subfield of ECG data analysis which involves the digitization of ECG recorded on paper (Lence et al., 2023), through devices such as Biopac modules (Maheshkumar et al., 2016). We have no experience with these methods and therefore we provided no recommendations on this specific ECG theme.

Although ECG computational analysis has become fundamental in cardiac research, it is sometimes carried out in the wrong way, with researchers making mistakes that lead to misleading results. Here, we presented a list of guidelines for the electrocardiogram signal processing to avoid common mistakes and wrong practices we noticed in several studies in the past.

Our list of quick tips, although partial, can serve as a starting point for good practices for researchers performing this kind of computational analysis. Although we designed these recommendations for apprentices and beginners, we believe they should be followed by experts, too. Complying with these guidelines can lead to better, more robust, and more reliable scientific results, that in turn can mean a better understanding of how patients’ hearts work, and therefore can lead to better therapies and better clinical outcomes.

List of abbreviations

CC Creative Commons

CSV comma-separated values

ECG electrocardiogram

FIR finite impulse response

HRV Heart Rate Variability

Hz hertz

IOR in-band to out-of-band spectral power ratio

MAE mean absolute error

MCC Matthews correlation coefficient

MAPE mean absolute percentage error

MSE mean square error

PYPL PopularitY of Programming Language

R-squared R2 coefficient of determination

RMSE root mean square error

RNA Ribonucleic acid

RNA-seq RNA-sequencing

SMAPE symmetric mean absolute percentage error

SQIs Signal Quality Indices.

Additional Information and Declarations

Competing Interests

Author Contributions

Data Availability

Davide Chicco is an Academic Editor for PeerJ Computer Science.

Davide Chicco conceived and designed the experiments, performed the experiments, analyzed the data, performed the computation work, prepared figures and/or tables, authored or reviewed drafts of the article, designed and conceived the quick tips, and approved the final draft.

Angeliki-Ilektra Karaiskou performed the experiments, analyzed the data, performed the computation work, prepared figures and/or tables, designed and conceived the quick tips, and approved the final draft.

Maarten De Vos conceived and designed the experiments, performed the experiments, performed the computation work, authored or reviewed drafts of the article, designed and conceived the quick tips, and approved the final draft.

The following information was supplied regarding data availability:

This is a literature review.

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
