# Peer review of "Ten quick tips for electrocardiogram (ECG) signal processing"

_PeerJ Computer Science, doi:10.7717/peerj-cs.2295_

## Round 0.1 · original submission · Major Revisions

The authors must respond to all observations of reviewers and to improve the article correspondingly.

Reviewer 1 ·

Basic reporting

It is appropriate.

Experimental design

It is appropriate but proper explanation is required.

Validity of the findings

(1) What are the novelties in the proposed work?
(2) What are the possible applications of the proposed work?
(3) What is the performance evaluating parameters of the proposed work?
(4) Manuscript needs inclusion of the necessary flow charts in the manuscript.
(5) Explain the performance evaluating parameters which were considered in the past literature and claim the novelty of the work.
(6) Major section on ECG Artifacts is required.
(7) As paper is based on ECG signal processing, following papers may be added-
(a) Adaptive autoregressive modeling based ecg signal analysis for health monitoring
(b) ECG Signal Analysis based on the Spectrogram and Spider Monkey Optimisation Technique
(c) Pre-Processing Based ECG Signal Analysis Using Emerging Tools
(d) Wavelet transform and vector machines as emerging tools for computational medicine
(e) A Novel FrWT Based Arrhythmia Detection in ECG Signal Using YWARA and PCA
(f) A review of different ECG classification/detection techniques for improved medical applications
(g) A Simplistic and Novel Technique for ECG Signal Pre-Processing
(h) An efficient AR modelling-based electrocardiogram signal analysis for health informatics
(i) Application of chaos theory for arrhythmia detection in pathological databases

Additional comments

No

Cite this review as

·

Basic reporting

Regarding the manuscript “Ten quick tips for electrocardiogram (ECG) signal processing” authored by Davide Chicco, Angeliki-Ilektra Karaiskou, and Maarten De Vos


• Manuscript is clear and English is okay but the thing is manuscript is written using word like you/your throughout which authors should avoid. Starting from line 55, 60, 62, 63, 65, 67, 71………………………… and so on
Academic writing often requires us to avoid first-person point of view in favor of third-person point of view, which can be more objective and convincing.
• Intro and Literature are well incorporated and are relevant.

Experimental design

• Rigorous investigation performed to a high technical & ethical standard.
• Methods described with sufficient detail & information to replicate.
• But there is one major concern from the reviewer:
Authors are explaining 12 lead configuration of ECG acquisition with support of figures which mostly reported in the printed form on paper. Additionally, Authors are talking about computational study which require data in computer accessible form. Although authors incorporated few publicly available datasets too. But the concern is still there regarding explained data acquisition methods.
Authors may additionally incorporate such data acquisition methods which gives the ECG data in computer interfaceable form (Such as BIOPAC or others which is 3-lead 2/4 channel computer interfaceable device).

Validity of the findings

• Authors have written a review which is better collected information from various sources and is already available so nothing unique.
• As a review article, conclusion meet future directions.

Additional comments

• Line 14: Use short form i.e ECG
Use ECG only not Electrocardiogram/ Electrocardiogram (ECG) as authors already used it once in abstract section and in list of Abbreviation. It occurs at multiple locations such as. Line no, 26, 47, 74, 86, 101, 113. 137, 161, 166, 176, 177, 183, 200, 207, 208, 224, 278, 287, 324, 330.
• Line 26: “and should not be confused with the electroencephalogram (EEG)”
Needless, Re-frame sentence or delete the sentence.
Line 38-43, 48: Avoid using word like 'I' or 'We'. Re-frame sentences. Academic writing often requires us to avoid first-person point of view in favor of third-person point of view, which can be more objective and convincing.
• Line 52: “Such as P wave, Q wave”: Incorporate P wave before Q wave
• Line 54: Line 55 should be with line 54 and not a separate para.
• Figure 1 caption: J-point not shown in Figure 1.
• Line 122: Next line should be together. Not with new para.
• Line 154: Ignore using 'to' multiple times.
• Line 237: Name those performance matrixes.
• Line 375: List of Abbreviation
Better to write abbreviation “Capitalise with each word” or “all small case’ but not mixed type.
• Reference:
References are not in common format. Such as in Ref 1, authors have used pages but not in other ref. Cross check all the references as per the journal references format.

Cite this review as

---

## Round 0.2 · accepted · Accept

The authors responded to all observations off reviewers. The article is well improved.